# Absolute Depth Measurement Using Multiphase Normalized Cross-Correlation for Precise Optical Profilometry

**DOI:** 10.3390/s19214683

**Published:** 2019-10-28

**Authors:** Duc-Hieu Duong, Chin-Sheng Chen, Liang-Chia Chen

**Affiliations:** 1Graduate Institute of Automation Technology, College of Mechanical & Electrical Engineering, No. 1, Section 3, National Taipei University of Technology, Zhong-Xiao E. Rd, Da’an District, Taipei City 10608, Taiwan; duongduchieu85@gmail.com (D.-H.D.); saint@mail.ntut.edu.tw (C.-S.C.); 2Department of Mechanical Engineering, National Taiwan University, No. 1, Section 4, Roosevelt Rd, Da’an District, Taipei City 10617, Taiwan

**Keywords:** fringe projection profilometry, temporal phase unwrapping, 3D measurement, multifrequency phase shifting, measurement uncertainty

## Abstract

In a multifrequency phase-shifting (MFPS) algorithm, the temporal phase unwrapping algorithm can extend the unambiguous phase range by transforming the measurement range from a short fringe pitch into an extended synthetic pitch of two different frequencies. However, this undesirably amplifies the uncertainty of measurement, with each single-frequency phase map retaining its measurement uncertainty, which is carried over to the final unwrapped phase maps in fringe-order calculations. This article analyzes possible causes and proposes a new absolute depth measurement algorithm to minimize the propagation of measurement uncertainty. Developed from normalized cross-correlation (NCC), the proposed algorithm can minimize wrong fringe-order calculations in the MFPS algorithm. The experimental results demonstrated that the proposed measurement method could effectively calibrate the wrong fringe order. Moreover, some extremely low signal-to-noise ratio (SNR) regions of a captured image could be correctly reconstructed (for surface profiles). The present findings confirmed measurement precision at one standard deviation below 5.4 µm, with an absolute distance measurement of 16 mm. The measurement accuracy of the absolute depth could be significantly improved from an unacceptable level of measured errors down to 0.5% of the overall measuring range. Additionally, the proposed algorithm was capable of extracting the absolute phase map in other optical measurement applications, such as distance measurements using interferometry.

## 1. Introduction

Precise absolute depth measurement remains one of the most important challenges in 3D surface profilometry because most 3D shape-measuring systems trade accuracy for depth range. With a significant increase in surface discontinuity, such as step height, measurement accuracy and precision cannot be guaranteed to satisfy engineering requirements. One of the most popular 3D measurement methods in industries is structured light projection, in which a structured light is projected onto an object’s surface to obtain the optical path difference (OPD) of the detected object point. The OPD is then captured by an imaging sensor in a triangular optical configuration. The deformed structured pattern can be further decoded to extract the OPD using phase-shifting or other spatial wrapping algorithms. Among the existing fringe projection techniques, phase-shifting profilometry [1] and Fourier-transform profilometry [2] are the two most popular approaches in determining the detected OPD through phase extraction processes. These two techniques estimate phase distribution using the arctangent function in phase wrapping. The estimated phase ranges between −π and π; hence, phase discontinuity is limited by the harmonic period of 2π. Phase unwrapping is thus required to eliminate the 2π phase discontinuities. Most phase-unwrapping approaches, including spatial phase unwrapping, involve comparing the phase values of neighboring pixels along the unwrapping paths [3,4,5,6,7,8,9,10,11,12]. The measured noise or surface discontinuity of the object shape can induce unwrapping errors, which are accumulated along unwrapping paths [13,14].

To resolve the problem of phase ambiguity, temporal phase unwrapping was developed by Saldner and Huntley [15]. This approach is derived from the phase value of neighboring pixels in the time domain, not in the spatial domain. The main idea is to combine additional coded patterns or several phase maps to extract the fringe order. The calculated fringe order is employed to unwrap the phase map. The absolute phase of each pixel is unwrapped independently along the time axis. Consequently, noisy pixels do not accumulate or infect other pixels in the unwrapping process.

This article focuses on the multifrequency phase-shifting (MFPS) technique, which is the most widely used in industrial applications. Although this technique can avoid phase ambiguity, it does have a critical drawback. The MFPS technique uses a combination of several phase maps with different fringe pitches; hence, measurement uncertainty is amplified and affects the final result. Each phase map of a single frequency has phase noises, which will lead to 3D measurement errors. An example of measurement uncertainty propagation after fringe-order calculation is illustrated in Figure 1, which also compares results between temporal phase unwrapping (multifrequency) and spatial phase unwrapping (Goldstein). Figure 1a shows an unwrapped phase map and its cross-section, which was obtained using temporal phase unwrapping. As can be seen, there is a jump point on the continuous surface caused by the wrong fringe order. However, this jump point disappears in the same pixel location in the unwrapped phase map shown in Figure 1b. Due to the measurement uncertainty of that pixel being amplified in fringe-order calculations, MFPS yields a wrong fringe order. The measurement uncertainty propagation of single- and multifrequency phase shifting techniques will be analyzed and modeled in Section 3.

In the MFPS technique, fringe order is calculated using the synthetic fringe pitch. The measurable range is extended after a combination of different pitches. The uncertainty of the synthetic pitch is also amplified in fringe-order calculations. This amplified uncertainty affects the 3D measurement results from fringe-order calculations. To minimize the amplification of measurement uncertainty, a new absolute height calibration technique is proposed. The calibrated absolute height will be employed to extract the fringe order with minimized uncertainty.

The remainder of this article is organized as follows. Section 2 reviews the fundamental theories of existing phase-shifting algorithms and measurement error. Critical analyses on measurement uncertainty propagation are then performed in Section 3, followed by the proposed methodology being put forward to address core research issues. Experimental results and analyses are presented in Section 4 to demonstrate the feasibility of the proposed measurement method in minimizing measurement uncertainty caused by the MFPS technique. Finally, Section 5 contains the conclusion.

## 2. Literature Review

### 2.1. Evolution from Single to Multifrequency Fringe Projection

In fringe projection profilometry (FPP), the distortion of fringe is estimated using a phase distribution map. Many techniques can extract the phase distribution of distorted fringes: the best-known ones are phase-shifting profilometry (PSP), wavelet-transform profilometry (WTP) [16,17], and Fourier-transform profilometry (FTP). PSP is widely used in industrial applications and many other fields because of its accuracy and efficiency. This technique can effectively minimize the interference of ambient light and surface reflectivity. Multiple sinusoidal fringe patterns with different phase shifts are projected onto the object’s surface. This distorted fringe distribution can be described using Equation (1):(1)$$I_{i}\left(x,y\right)=A\left(x,y\right)+M\left(x,y\right)\cos\left(\phi^{w}-\frac{2\pi i}{N}\right)$$
where $A\left(x,y\right)$ represents ambient light from the environment, $M\left(x,y\right)$ denotes intensity modulation; i is the phase-shifted index, i=0,1,…,N−1; and $\phi^{w}\left(x,y\right)$ is the phase distribution map of distorted fringes, also called the wrapped phase map.

The wrapped phase map can be extracted by resolving at least three different phase-shifted equations (Equation (1)). The wrapped phase map is calculated using Equation (2):(2)$$\phi^{w}\left(x,y\right)=\arctan\left(\frac{\sum_{i=0}^{N-1}I_{i}\left(x,y\right)\sin\left(\frac{2\pi i}{N}\right)}{\sum_{i=0}^{N-1}I_{i}\left(x,y\right)\cos\left(\frac{2\pi i}{N}\right)}\right)$$

Figure 2 illustrates the setup of the most common FPP system in industrial applications. In FPP, the optical axis of projection and the observer must construct a triangle optical configuration, and the object’s surface height is calculated using the triangulation rule. The object’s height is computed according to the relationship between distorted phase distribution on an object’s surface and a reference plane. The final object height can be extracted using Equation (3):(3)$$Z\left(x,y\right)=\frac{P\left(x,y\right)\Delta \phi^{u}\left(x,y\right)}{2\pi \left(\tan\alpha +\tan\beta \right)}$$
where $Z\left(x,y\right)$ is the difference in height between measured points located on the object surface and the reference grid; $P\left(x,y\right)$ is the pitch of the reference grid; α  is the projection angle; β is the observation angle; and $\Delta \phi^{u}\left(x,y\right)$ is the unwrapped phase difference.

When the multifrequency fringe projection is employed, the wrapped phase can be unwrapped from the coarsest to the finest fringe pitch [15]. In the following equation, the two wrapped phase maps are denoted as $\phi_{l}^{w}$ and $\phi_{h}^{w}$, while the unwrapped phase maps are denoted as $\phi_{l}^{u}$ and $\phi_{h}^{u}$, with a fringe pitch of $P_{l}$ and $P_{h}$, respectively ($P_{h}<P_{l}$; subscripts “*h*” and “*l*” denote “high frequency” and “low frequency”, respectively). The relation between the unwrapped phase map and the wrapped phase map of high frequency can be expressed by Equation (4):(4)$$\phi_{h}^{u}\left(x,y\right)=\phi_{h}^{w}\left(x,y\right)+2\pi n_{h}\left(x,y\right)$$

Similarly, the relation between the unwrapped phase map and the low-frequency wrapped phase map can be formulated as Equation (5):(5)$$\phi_{l}^{u}\left(x,y\right)=\phi_{l}^{w}\left(x,y\right)+2\pi n_{l}\left(x,y\right)$$

The relationship between the low- and high-frequency fringe pitches can be expressed as Equation (6):(6)$$\frac{\phi_{h}^{u}\left(x,y\right)}{\phi_{l}^{u}\left(x,y\right)}=\frac{P_{l}}{P_{h}}$$

The ratio, $\frac{\phi_{h}^{u}\left(x,y\right)}{\phi_{l}^{u}\left(x,y\right)}$, of the projected fringe pitches can be used as a conversion between the unwrapped phases of high- and low-frequency fringes.

There are four main strategies for temporal phase unwrapping [18,19,20,21,22,23,24,25]: hierarchical (unit frequency), numerical (theoretical), heterodyne (multifrequency), and others. In the hierarchical strategy, the unit fringe frequency is projected to cover the entire measured field-of-view (FOV). The wrapped phase map of the unit fringe does not need any phase unwrapping. The other phase maps are extracted one by one from their previous unwrapped phase maps according to the relationships between their fringe pitches. Numerical (theoretical) temporal phase unwrapping relies on a suitably chosen fringe pitch, $P_{l}$ and $P_{h}$. A unique set of pairs $\left(\phi_{l}^{w},\;\phi_{h}^{w}\right)$ can be obtained along with the absolute phase axis. Similarly, in general, each set of wrapped phase values of *K*-frequencies is unique [26]. A lookup table is created to store the relation between the fringe order and the unique wrapped phase set. Therefore, the fringe orders of these phase maps ($n_{l}$ and $n_{h}$) can be extracted by estimating their wrapped phase values. The height of the object surface can be obtained by calculating the exponential sums of wrapped phases [27]. Furthermore, in the multifrequency strategy, one of nature’s effects, synthetization, is applied. The combination of more than one frequency to form an expanded frequency is called synthetization. By employing this effect, several fringe pitches are projected to extend the unambiguous phase range to the synthetic pitch at the beat fringe pitch of two close frequencies. As mentioned by Saldner and Huntley [15], the synthetic pitch can be calculated using Equation (7):(7)$$P_{syn}=\frac{P_{l}P_{h}}{P_{l}-P_{h}}$$

The synthetic phase map is generated from the wrapped phase difference between low- and high-frequency phase maps and can be expressed as Equation (8):(8)$$\phi_{syn}^{w}\left(x,y\right)=\phi_{h}^{w}\left(x,y\right)-\phi_{l}^{w}\left(x,y\right)$$

The final unwrapped phase map will be extracted by applying Equations (4)–(6) when $\phi_{syn}^{w}\left(x,y\right)$ is replaced by $\phi_{l}^{w}\left(x,y\right)$. Other temporal phase-unwrapping approaches are to extract the absolute phase with the aid of special structured patterns such as Gray code or speckle [28,29]. In these approaches, the fringe order is encoded by serial Gray-code patterns. *N* patterns can only encode $2^{N}$ fringe orders, thus prolonging data acquisition. Furthermore, the edges of captured binary patterns blur due to the defocusing of the optical system. Incorrect absolute phases frequently appear on the object’s sharp edge or on the boundary between adjacent binary fringe regions.

### 2.2. Measurement Errors Incurred in PSP

To understand how various measurement errors encountered in PSP impact phase measurement accuracy, the measurement uncertainties in PSP were analyzed and modeled. In the analysis, the sensor noise variation, $\sigma^{2}$, was assumed to fit the Gaussian distribution. Most common sensors have two major types of noises: shot noise (random) and dark current noise (coming from thermal energy). When the ratio between the sensor noise and the intensity signal is reasonably small, the effect of noise on the measured phase can be ignored. The noise effect can be considered to be the first-order approximation of phase error variation [23,30]. In general, the phase error of the wrapped phase includes systematic error and random error. In FPP systems that employ structured-light projectors, nonsinusoidal error can dominate as the major error. Nonsinusoidal phase errors can be satisfactorily modeled as a periodic sinusoidal function with multiple spatial frequencies of a multiple number of phase-shifting steps [31,32,33,34,35]:(9)$$\sigma_{\phi }^{2}=\overset{N-1}{\underset{i=0}{\sum }}\left[{\left(\frac{\partial \phi }{\partial I_{i}}\right)}^{2}\sigma^{2}\right]$$

Phase shifting is independently evaluated on a single pixel; hence, image coordinates $\left(x,y\right)$ are ignored in Equation (9). The quantitative phase error is calculated by taking the first derivative of the phase calculation model (Equation (2)). The main difference between single-frequency and multifrequency phase shifting is that different fringe pitches are captured to archive unwrapped phase maps in multifrequency phase shifting. The fringe order of high-frequency phase maps, $n_{h}\left(x,y\right)$, can be calculated using the unwrapped phase of a synthetic-frequency phase map [23] and can be expressed as Equation (10):(10)$$n_{h}\left(x,y\right)=INT\left[\frac{\left(\frac{P_{syn}}{P_{h}}\right)\phi_{syn}^{w}\left(x,y\right)-\phi_{h}^{w}\left(x,y\right)}{2\pi }\right]$$
where $\phi_{h}^{w}\left(x,y\right)$ and $\phi_{syn}^{w}$ are the high and synthetic wrapped phases, respectively; $P_{h},\;P_{syn}$ are the fringe pitches of high and synthetic frequencies of projected fringe projection, respectively; and *INT* is the integer operation (because fringe order must be an integer number).

It is important to note that a fringe-order error will occur if the *INT* operation in Equation (12) fails. Consequently, fringe-order error can be quantified using Equations (11) and (12):(11)$$\left[\frac{\left(\frac{P_{syn}}{P_{h}}\right)\phi_{syn}^{w}\left(x,y\right)-\phi_{h}^{w}\left(x,y\right)}{2\pi }\right]>0.5$$
where Equation (11) can be further simplified as
(12)$$\left|P_{syn}\phi_{syn}^{w}\left(x,y\right)-P_{h}\phi_{h}^{w}\left(x,y\right)\right|>\pi P_{h}$$

From the above derivation of the fringe-order calculation error, it can be concluded that the measurement uncertainty of multifrequency fringe projection is excessive and cannot be ignored. This study analyzes the measurement uncertainty and proposes a normalized cross-correlation (NCC)-based calibration method to effectively reduce the uncertainty of fringe order, thus enhancing the measurement accuracy of multifrequency fringe projection.

## 3. Proposed NCC-Based Methodology

### 3.1. Measurement Uncertainty Analysis

The unambiguous phase range is extended by transferring the measurement range from the short to the synthetic fringe pitch. However, each single-frequency phase map retains measurement uncertainty, which is seen in unwrapped high-frequency phase maps with fringe-order calculations. In addition, the synthetic fringe pitch is amplified after the combination of different fringe frequencies. This process also amplifies the measurement uncertainty of involved single-frequency phase maps. In this paper, the phase noise model was derived by following the phase noise model of temporal phase-unwrapping algorithms, which was proposed by Zuo et al. in 2016 [23]. Measurement uncertainty propagation with fringe-order calculations can be computed using Equation (13):(13)$$\sigma_{n}^{2}=\left(\frac{P_{l}^{2}}{{\left(P_{l}-P_{h}\right)}^{2}}+1\right)\sigma_{\phi_{h}}^{2}+\frac{P_{l}^{2}}{{\left(P_{l}-P_{h}\right)}^{2}}\sigma_{\phi_{l}}^{2},$$
where $\sigma_{\phi_{l}},\sigma_{\phi_{h}}$ denote the uncertainty of low- and high-frequency fringe pitches, respectively.

The proposed method aims to eliminate the amplified ratio $\left(\frac{P_{h}^{2}}{{\left(P_{l}-P_{h}\right)}^{2}},\frac{P_{l}^{2}}{{\left(P_{l}-P_{h}\right)}^{2}}\right)$ of measurement uncertainty after fringe synthetization. Figure 3 illustrates the conceptual illustration of the proposed NCC-based method, in which the measured phase set on an arbitrary object surface is accurately correlated and mapped between the reference phase set in the database and the calibrated surface depth. The quantitative matching score of the measured phase map with each reference single-frequency phase map in the database is estimated using a normalized cross-correlation (NCC) algorithm. By doing so, it is assumed that the uncertainty of every single-frequency phase map is constrained and will not be amplified in fringe synthetization, in which the amplified ratio in fringe-order calculations does not affect the final unwrapped phase map. To prove this, the measurement uncertainty in the proposed method is analyzed below.

Assuming that the stored reference phase $\phi_{dbk}$ in the database and the distorted phase $\phi_{mk}$ retain uncertainty ($\sigma_{\phi_{dbk}}$ and $\sigma_{\phi_{mk}}$, respectively), with k=1,2,…,K denoting a different frequency index, the NCC-based method is employed to calculate the matching score between the measured phase value and the reference phase value in the database using Equation (14):(14)$$NCC=\frac{\sum_{k=1}^{K}\omega_{mk}.\omega_{dbk}}{\sqrt{\sum_{k=1}^{K}\omega_{mk}{}^{2}\sum_{k=1}^{K}\omega_{dbk}{}^{2}}}$$
where $\phi_{mk}$ and $\phi_{dbk}$ are the phase values of the measurement and the stored database of frequency order k. In addition, $\omega_{mk}$ can be further obtained using Equation (15),
(15)$$\omega_{mk}=\phi_{mk}-\frac{1}{K}\sum_{k=1}^{K}\phi_{mk}$$
and $\omega_{dbk}$ is acquired using Equation (16),
(16)$$\omega_{dbk}=\phi_{dbk}-\frac{1}{K}\sum_{k=1}^{K}\phi_{dbk}$$

Similarly, the uncertainty is propagated from the wrapped phase of *K* different frequencies to the matching score after NCC calculation. The final propagated measurement uncertainty is expressed as
(17)$$\sigma_{NCC}{}^{2}=\sum_{k=1}^{K}\left[{\left(\frac{\partial NCC}{\partial \phi_{mk}}\right)}^{2}\sigma_{\phi_{mk}}{}^{2}\right]+\sum_{k=1}^{K}\left[{\left(\frac{\partial NCC}{\partial \phi_{dbk}}\right)}^{2}\sigma_{\phi_{dbk}}{}^{2}\right]$$

Equation (17) can be further expressed as
(18)$$\begin{array}{l} \sigma_{NCC}{}^{2}=\sum_{k=1}^{K}\left\{{\left[\left(1-\frac{1}{K}\right)\frac{\omega_{mk}\sum_{k=1}^{K}\omega_{mk}\omega_{dbk}\sum_{k=1}^{K}\omega_{dbk}^{2}-\omega_{dbk}\sum_{k=1}^{K}\omega_{mk}^{2}\sum_{k=1}^{K}\omega_{dbk}^{2}}{{\left(\sum_{k=1}^{K}\omega_{mk}^{2}\sum_{k=1}^{K}\omega_{dbk}^{2}\right)}^{\frac{3}{2}}}\right]}^{2}\sigma_{\phi_{mk}}{}^{2}\right\}\\ +\sum_{k=1}^{K}\left\{{\left[\left(1-\frac{1}{K}\right)\frac{\omega_{dbk}\sum_{k=1}^{K}\omega_{mk}\omega_{dbk}\sum_{k=1}^{K}\omega_{mk}^{2}-\omega_{mk}\sum_{k=1}^{K}\omega_{mk}^{2}\sum_{k=1}^{K}\omega_{dbk}^{2}}{{\left(\sum_{k=1}^{K}\omega_{mk}^{2}\sum_{k=1}^{K}\omega_{dbk}^{2}\right)}^{\frac{3}{2}}}\right]}^{2}\sigma_{\phi_{dbk}}{}^{2}\right\} \end{array}$$

With the inequality of Cauchy–Bunyakovsky–Schwarz [36], Equation (18) can be simplified as
(19)$$\sigma_{NCC}{}^{2}\leq \sum_{k=1}^{K}\left\{{\left(1-\frac{1}{K}\right)}^{2}{\left(\frac{\omega_{mk}\sum_{k=1}^{K}\omega_{dbk}^{2}}{{\left(\sum_{k=1}^{K}\omega_{mk}\omega_{dbk}\right)}^{2}}-\frac{1}{\sqrt{\left(K.\sum_{k=1}^{K}\frac{\omega_{mk}^{2}}{\omega_{dbk}^{2}}\right)}}\right)}^{2}\sigma_{\phi_{mk}}{}^{2}\right\}\;+\sum_{k=1}^{K}\left\{{\left(1-\frac{1}{K}\right)}^{2}{\left(\frac{\omega_{dbk}\sum_{k=1}^{K}\omega_{mk}^{2}}{{\left(\sum_{k=1}^{K}\omega_{mk}\omega_{dbk}\right)}^{2}}-\frac{1}{\sqrt{\left(K.\sum_{k=1}^{K}\frac{\omega_{dbk}^{2}}{\omega_{mk}^{2}}\right)}}\right)}^{2}\sigma_{\phi_{dbk}}{}^{2}\right\}$$
where $0\leq \phi_{mk}\ll 2\pi$ and $0\leq \phi_{dbk}\ll 2\pi$.

From Equations (15) and (16), the limits of $\omega_{mk}$ and $\omega_{dbk}$ can be determined as
(20)$$-\pi \leq \omega_{mk}\leq \pi$$
(21)$$-\pi \leq \omega_{dbk}\leq \pi$$

From the above equations, the following inequality can be expressed
(22)$$\frac{1}{\sqrt{\left(K.\sum_{k=1}^{K}\frac{\omega_{mk}^{2}}{\omega_{dbk}^{2}}\right)}}\geq 0$$

By substituting Equation (22) into the first element in the bracket of Equation (19), Equation (23) can be formed:(23)$$\frac{\omega_{mk}\sum_{k=1}^{2}\omega_{dbk}^{2}}{{\left(\sum_{k=1}^{2}\omega_{mk}\omega_{dbk}\right)}^{2}}-\frac{1}{\sqrt{\left(K.\sum_{k=1}^{K}\frac{\omega_{mk}^{2}}{\omega_{dbk}^{2}}\right)}}\leq \frac{\omega_{mk}\sum_{k=1}^{2}\omega_{dbk}^{2}}{{\left(\sum_{k=1}^{2}\omega_{mk}\omega_{dbk}\right)}^{2}}$$

In addition, the righthand side of Equation (19) can reach its maximum when $\omega_{mk}=\pi$ and $\omega_{dbk}=\pi$:
(24)$$\frac{\omega_{mk}\sum_{k=1}^{K}\omega_{dbk}^{2}}{{\left(\sum_{k=1}^{K}\omega_{mk}\omega_{dbk}\right)}^{2}}\leq \frac{\pi \sum_{k=1}^{K}\pi^{2}}{{\left(\sum_{k=1}^{K}\pi .\pi \right)}^{2}}=\frac{K\pi^{3}}{K^{2}\pi^{4}}=\frac{1}{K\pi }$$
(25)$$\frac{\omega_{dbk}\sum_{k=1}^{2}\omega_{mk}^{2}}{{\left(\sum_{k=1}^{2}\omega_{mk}\omega_{dbk}\right)}^{2}}\leq \frac{\pi \sum_{k=1}^{2}\pi^{2}}{{\left(\sum_{k=1}^{2}\pi .\pi \right)}^{2}}=\frac{K\pi^{3}}{K^{2}\pi^{4}}=\frac{1}{K\pi }$$

By further substituting Equations (24) and (25) into Equation (19), the final propagated phase uncertainty can be expressed as
(26)$$\sigma_{NCC}{}^{2}\leq \sum_{k=1}^{K}\left\{{\left(1-\frac{1}{K}\right)}^{2}{\left(\frac{1}{K\pi }\right)}^{2}\sigma_{\phi_{mk}}{}^{2}\right\}+\sum_{k=1}^{K}\left\{{\left(1-\frac{1}{K}\right)}^{2}{\left(\frac{1}{K\pi }\right)}^{2}\sigma_{\phi_{dbk}}{}^{2}\right\}$$

Equation (26) indicates clearly that the distorted phase uncertainty varies with the uncertainty of the wrapped phase $\sigma_{\phi_{mk}}$ and the reference phase in the database $\sigma_{\phi_{dbk}}$. In addition, the uncertainty of the reference phase in the database is much smaller than that of the distorted phase because the reference phase can be established using a precalibrated accurate reference height with minimized random measured noises, and its remaining measured uncertainty is again minimized by a 30-times time-averaging strategy. Since the measured random noise in the stored reference phase map of each single frequency is normally carried to the final calibrated absolute height, the potential random noises need to be minimized in the stored *K*-frequency phase maps before being saved to the reference database. To realize this, a median filter with a 3 × 3 mask (using a signal time-averaging process) can be employed to effectively minimize the random noise of *K*-frequency phase maps.

### 3.2. NCC-Based Method for Absolute Depth Measurement

The proposed method is generally applicable to any multifrequency fringe projection with two or more different frequencies. In a general FPP-based system, the reference plane is aligned perpendicularly to the optical axis of the imaging lens in front of the sensor and can be vertically shifted to determine a precise depth to be calibrated. To ensure good quality of the captured fringe image, it is important to keep the measurable range strictly within the overlapped depth-of-field (DOF) between the sensor lens and the projection lens in the FPP system. In the experimental system setup, the maximum measurable range is defined as *H* when the reference plane is shifted perpendicularly to the optical axis of the sensor lens for *Q* layers with a height pitch Δh in a single step. In Figure 4, the NCC-based absolute-depth measuring method is explicitly illustrated. In a multifrequency FPP, for each individual fringe projection, it is essential that the *N*-step phase-shifting image bucket is performed in a time-averaging manner multiple times (*M*) to minimize random phase errors (in which *M* is set at 30 or more). In general, phase noises may be introduced from lighting illumination, sensor noises, and other possible environmental sources. The phase maps of *K*-frequencies are calibrated and obtained from each individual fringe projection and are stored in the reference database, as shown in Figure 4. During the actual measuring task, when the same *K*-frequency sinusoidal fringes are sequentially projected onto a tested object with an arbitrary surface profile, PSP is performed for each individual fringe to measure and determine the *K*-frequency phase maps.

The proposed method works only when the uniqueness of the set of all wrapped phases in the multifrequency fringe projection has a phase value from 0 to 2π (which is reconstructed by the synthetic fringe). A pair of arbitrary phases, $\left(\phi_{l}^{u},\phi_{h}^{u}\right)$, can be employed to determine the corresponding absolute phase value, $\phi^{abs}=P_{l}\phi_{l}^{u}\left(x,y\right)=P_{h}\phi_{h}^{u}\left(x,y\right)$, and the following equation holds:(27)$$P_{l}\phi_{l}^{w}\left(x,y\right)+2\pi P_{l}n_{l}\left(x,y\right)=P_{h}\phi_{h}^{w}\left(x,y\right)+2\pi P_{h}n_{h}\left(x,y\right)$$

Equation (27) presents a one-to-one relation between $\phi_{l}^{w}$ and $\phi_{h}^{w}$ along the absolute phase axis. Consequently, the unique pairs of $\left(\phi_{l}^{w},\phi_{h}^{w}\right)$ can always be obtained when they propagate along the absolute phase axis. This relationship can be further proven and demonstrated using a one-to-one function or the injective function definition in Reference [37].

To demonstrate that each wrapped phase of the synthetic fringe is unique, an example with a fringe pitch of two different frequencies is considered, in which $P_{l}\;\mathrm{and}\;P_{h}$ are set at 4 and 5 units, respectively. According to Equation (7), the synthetic fringe pitch is at 20 units, in which there exist four fringes with spacing $P_{l}$ and five fringes with spacing $P_{h}$. Within the range of the synthetic pitch, it is important to point out that all of the phases are unambiguous and unique [38]. The concept of this idea is illustrated in Figure 5a.

Figure 5b shows the fractional phase $d_{f}$ (rescaled from 0 to 2π) as measured with two different fringe pitches when the absolute phase increases. Each combination of relative wrapped phases is unique until the synthetic fringe pitch reaches 20 units. However, the combinations of the wrapped phases $\left(\phi_{l}^{w},\;\phi_{h}^{w}\right)$ remain unique until the fractional phase equals 0 again. The above principle can also be applied when $d_{f}$ is rescaled according to the pixel resolution and an arbitrary number of different frequencies are applied.

Figure 6 illustrates the concept of the proposed absolute height calculation algorithm for every tested image pixel. A set of the measured phase values $\phi_{mk}$ of the *K*-frequency fringe projection is digitally correlated with the corresponding reference phases stored in the database, $\phi_{dbk}$, by calculating the NCC values between the measured phase set and the reference phase set. The absolute height of the tested object surface can be determined at a depth by finding the maximum matching score of the NCC. Full-field absolute-height surface profilometry can be achieved by repeating the same process for all of the other measured pixels.

Figure 7 presents a flowchart of the proposed NCC-based multiphase depth measurement algorithm. In a measuring task, a set of *K*-frequency phase-shifting images is acquired, in which the phase maps of every single frequency are calculated. The wrapped phase maps are employed to calibrate the absolute height using the proposed NCC-based method. The final unwrapped phase map can be reconstructed using the relationship in Equation (14), in which the absolute phase can be reconstructed accurately with the extracted fringe order $n\left(x,y\right)$ and the wrapped phase of the finest fringe pitch $\phi^{w}\left(x,y\right)$.

### 3.3. Fringe-Order Determination

When an absolute height is determined using the above method, an absolute phase map can be further reconstructed using the calculated fringe order and the wrapped phase map. The absolute height is employed to compute the fringe order for phase unwrapping. To achieve this, a phase-to-height calibration process is performed to accurately determine the mapping relationship between the absolute phase difference and the surface absolute depth. Equation (3) presents the relationship between the absolute phase and the surface absolute depth. Before transferring the absolute phase to the height, the phase discontinuity existing in the wrapped phase map has to be recovered by compensating for the missing fringe order, $n\left(x,y\right)$.

Figure 8 illustrates the proposed idea of how the missing fringe order $n\left(x,y\right)$ is related to the absolute height in a single pixel. The absolute height of a single pixel can be defined by its wrapped phase value and the fringe order incurred by the object’s absolute height (Equation (28)). The accurate mapping relationship between the fringe order $n\left(x,y\right)$, the wrapped phase $\phi^{w}\left(x,y\right)$, and the unwrapped phase $\phi^{u}\left(x,y\right)$ can be described by Equation (29). The unwrapped phase can be extracted using the calibrated absolute height and the relationship between the absolute phase and the reference phase $\phi_{ref}^{abs}\left(x,y\right)$. In addition, the relationship of the calibrated absolute height $h_{abs}\left(x,y\right)$ with the calibrated phase-to-height coefficient, $C_{z}$, and the unwrapped phase $\phi^{u}\left(x,y\right)$ can also be expressed by Equations (28) and (29), respectively. Thus, an accurate fringe order can be determined using the calibrated absolute height, wrapped and reference phase maps, as in Equation (30). Equations (28) and (29) are
(28)$$\phi^{u}\left(x,y\right)=\phi_{ref}^{abs}\left(x,y\right)\;+\;\frac{h_{abs}\left(x,y\right)}{C_{z}}$$
(29)$$\phi^{u}\left(x,y\right)=\phi^{w}\left(x,y\right)+2\pi n\left(x,y\right)$$

The fringe order can then be expressed:(30)$$n\left(x,y\right)=floor\left(\frac{\;\frac{h_{abs}\left(x,y\right)}{C_{z}}-\left[\phi^{w}\left(x,y\right)-\phi_{ref}^{abs}\left(x,y\right)\right]}{2\pi }\right)$$
where $floor\left(X\right)$ returns the largest integral value that is not greater than X.

The final absolute phase map is transferred to the object’s height using phase-to-height calibration coefficients, which are estimated in the system calibration process.

Using Equations (28) and (29), the fringe-order uncertainty of the proposed method can be further expressed as
(31)$$\sigma_{n}{}^{2}=\frac{\sigma_{NCC}{}^{2}}{C_{Z}^{2}}+\sigma_{\phi_{h}}{}^{2}$$

## 4. Experimental Results and Analyses

During the experimental measurements, a projector projected sinusoidal patterns onto an object surface, and distorted fringe patterns were acquired by a sensor (see Figure 9a). To build the database, the reference plane was aligned perpendicularly to the optical axis and shifted along the optical axis of the sensor lens. Each five-step phase-shifting image set was acquired 30 times. The 30 image sets were then averaged to minimize the random error from illumination and the sensor. In practice, the measurable range was *H* = 10 mm, and one shifted step of the reference plane was Δh=0.1 mm; hence, there were, in total, *Q* = 100 layers stored in the database. A high accurate ceramic plate (flatness and roughness <5 µm) was chosen to build the database for absolute height calculations.

To ensure the accuracy of this database, the reference plane was mounted onto a precalibrated accurate linear stage with a minimum incremental motion of 0.1 µm and an accuracy of 0.8 µm. The travel range of the linear stage was 25 mm, which covered the entire measurable range of the system. On the imaging side, a telecentric lens with a DOF of 16.7 mm was chosen. Figure 9b shows a practical system for fringe projection profilometry. The projected fringes were created by a projector, with the four pitches being 121, 110, 100, and 12 pixels of digital micromirror device (DMD). The first three fringe pitches were chosen as the optimized frequency to extend the measurable range of the measurement system. The highest fringe frequency with a fringe pitch of 12 pixels was chosen as the finest fringe pitch to produce the best measuring resolution of the experimental system.

Figure 10a–c shows an example of the images being captured on a hemisphere target, which was illuminated through projection of the largest fringe pitch (121 pixels) and the shortest fringe pitch (12 pixels) (four fringes (12, 100, 110, and 121) were applied). Figure 10d,e shows the wrapped phase maps of Figure 10b,c, respectively. The absolute height map calculated using the proposed algorithm is illustrated in Figure 10f. The 3D profile and its height cross-section are presented in Figure 10g,h. As can be seen in Figure 10, the measurement results showed that the absolute height of the hemisphere surface could be successfully reconstructed using the proposed NCC-based algorithm.

A step-height target was measured to evaluate the precision and accuracy of the proposed absolute height calculation technique. A step-height target was made of stainless steel through a strictly controlled grinding process. The parallelism, flatness, and step-heights of the target were measured using a Mitutoyo CMM for accurate estimation of the optical probe. Figure 11 presents a calculated fringe-order map of the precalibrated step-height target using MFPS and the proposed algorithm. Some wrong fringe-order pixels appeared near the border of the two measured step heights when the MFPS algorithm was applied (Figure 11a). In comparison, as can be seen in Figure 11b, the proposed algorithm could effectively correct the wrong fringe-order pixels shown in Figure 11a.

The final absolute phase map was extracted by combining the calculated fringe order and the wrapped phase of the finest fringe projection, thus achieving high depth resolution and absolute depth measurement simultaneously. To determine the measurement accuracy of the proposed method, tests were performed on a step-height target and on industrial parts, including a cell phone button, screws (shiny and black), and an electrical connector. The measured results obtained using MFPS and NCCMFPS were analyzed and compared.

Figure 12 illustrates the measurement results obtained using MFPS and NCCMFPS on a precalibrated accurate 2-mm step-height target. Figure 12a presents the 3D profile of the step-height target with two height deviations between three different planes. The step heights between Section 1, Section 2 and Section 3 were quantitatively evaluated for a measurement accuracy comparison between the two methods. The height cross-sections on column 256 for both MFPS and NCCMFPS are presented in Figure 12b. The blue and red curves represent the height cross-sections reconstructed using NCCMFPS and MFPS, respectively. A clear undesired jump around the indicated point can be seen in Figure 12b, which was reconstructed using MFPS.

To estimate the accuracy of the proposed method, the average step heights between Section 1, Section 2 and Section 3 in Figure 12b were calculated. Table 1 presents the measurement results for the same step-height target obtained using MFPS, NCCMFPS, and the Mitutoyo CMM. As can be seen, the absolute height measuring errors of the proposed method and the traditional MFPS method at pixel position 299 were 0.010 mm and 13.590 mm, respectively. The measurement results demonstrated a significant improvement in measurement accuracy for the absolute depth using the proposed method.

The absolute distances between the centers of the three different spheres were extracted using their measured point cloud. Figure 13a illustrates the measured 3D profile of the step-height target with three spheres mounted on top of it. Figure 13b illustrates the segmented point cloud and calculated sphere centers using a random sample consensus (RANSAC) fitting algorithm. To estimate the stability of the probe, the Euclidean distances between the centers of the spheres were measured 30 times under the same conditions. Table 2 shows that the standard deviation of the probe could reach 5.4 µm. Figure 13c,d illustrates two broken areas inside the red ellipses, in which the proposed NCCMFPS method had a much higher signal-to-noise ratio (SNR) reconstruction around the edges.

When testing the feasibility of the proposed method, a cell phone button was chosen to represent an arbitrary shape. Figure 14 shows the measured results for the cell phone target and a comparison between MFPS and NCCMFPS. Figure 14g,h shows the 3D profile of the cell phone target measured using MFPS and NCCMFPS, respectively. Two incorrect regions caused by wrong fringe-order calculations (from MFPS) in low SNR regions are marked inside the red ellipses in Figure 14g. These regions were calibrated to their correct positions using NCCMFPS. Some noise areas remained because the SNR of sinusoidal fringe in these areas could not be well detected by a sensor, but they were minimized as much as possible.

The most challenging problem using the FPP technique was nonuniform reflectivity and the high curvature of the measured object’s surface, which appears in Figure 15a. Several industrial parts were mixed together, with shiny and black screws giving large reflectivity variations; a white plastic electrical connector representing a good SNR target; and a copper bolt representing a target with a medium level of SNR but with high curvature. Figure 15g is a fringe-order map of mixed industrial parts measured using MFPS. There were several broken regions on the shiny screw and in the corner areas of the white plastic connector. The wrong fringe order in these regions propagated to the final measured 3D profile, as presented inside the yellow ellipses in Figure 15i. These broken areas were calibrated to their correct positions using NCCMFPS, as shown in Figure 15j. Taken together, the experimental results demonstrated that the proposed measurement method (NCCMFPS) could effectively calibrate fringe-order errors caused by the MFPS algorithm.

## 5. Conclusions

In this article, new absolute height and fringe-order calculations developed from multiphase NCC, which effectively minimizes fringe-order error in MFPS, was proposed. Measurement uncertainty was successfully minimized by eliminating amplified ratios in the fringe pitch synchronization. The experimental results showed that the proposed measurement method could effectively calibrate fringe-order errors caused by the MFPS technique. The accuracy of the proposed approach was estimated using absolute distances in the center space of three spheres. The measurement results showed that the measurement accuracy of absolute depth could be significantly improved from an unacceptable level of measured errors down to 0.5% of the overall measuring range. Additionally, the proposed algorithm was also capable of extracting the absolute phase map in many different applications using multiphase maps for measurements, such as structured light projection, interferometry, and optical inspection.

## Figures and Tables

**Figure 1 sensors-19-04683-f001:**
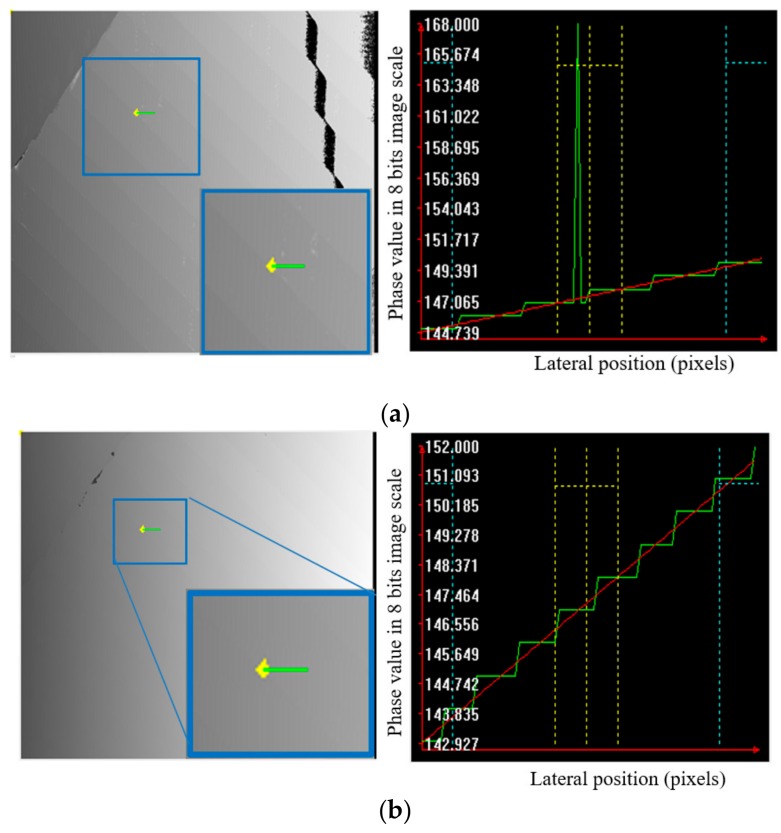
Unwrapped phase map and its cross-section measured by a fringe pitch of 12 pixels on a low-reflectivity step-height target using (**a**) temporal phase unwrapping (multifrequency) and (**b**) spatial phase unwrapping (Goldstein [5]).

**Figure 2 sensors-19-04683-f002:**
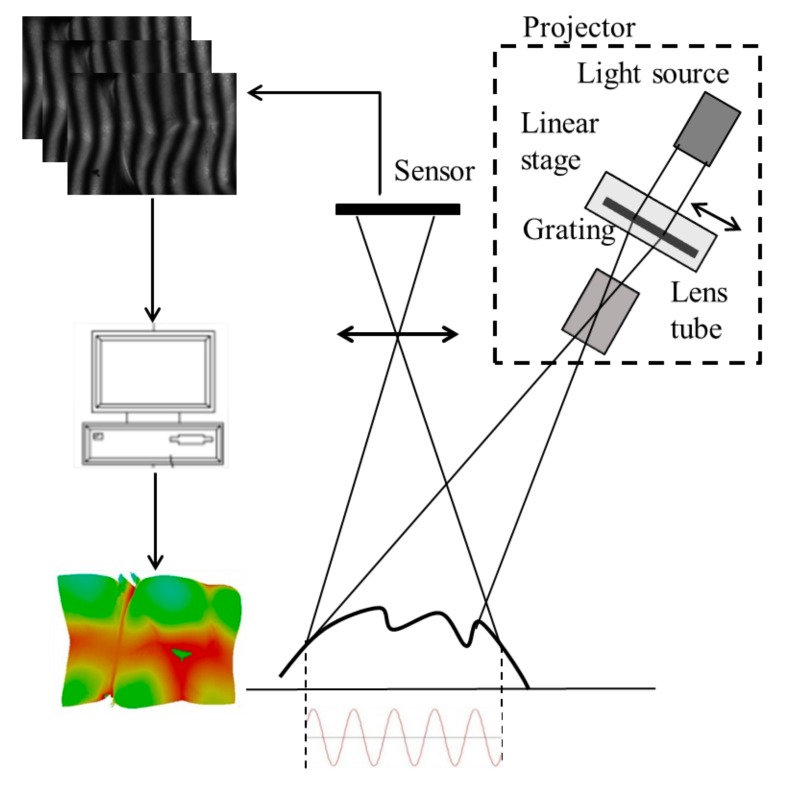
System setup of fringe projection profilometry.

**Figure 3 sensors-19-04683-f003:**
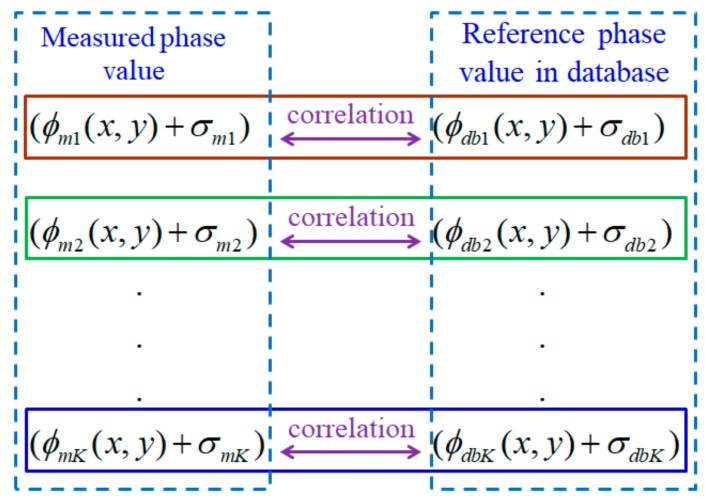
Concept illustration of the correlation between multifrequency measured phases and the reference phases of *K*-frequencies stored in the database.

**Figure 4 sensors-19-04683-f004:**
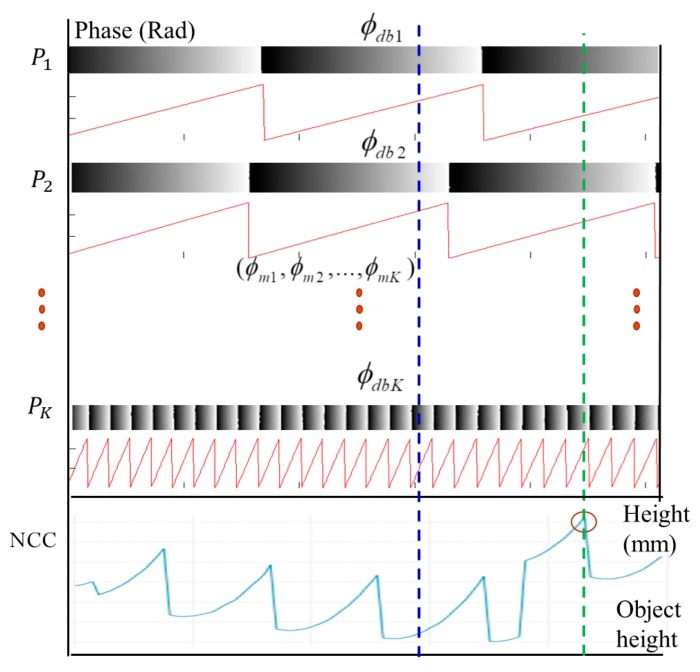
Illustration of the proposed absolute height calibration algorithm for a single measuring pixel.

**Figure 5 sensors-19-04683-f005:**
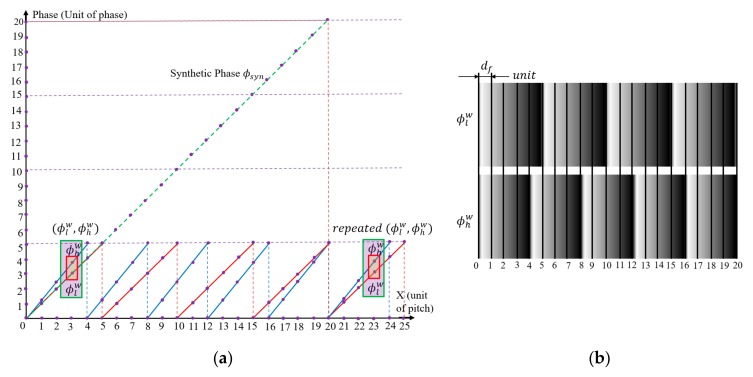
Basic idea of numerical (theoretical) temporal phase unwrapping: (**a**) fraction of wrapped phases with $P_{l\;}(=5$) units and $P_{h}(=4$ ) units; (**b**) corresponding wrapped phases, with integer steps indicated by vertical lines.

**Figure 6 sensors-19-04683-f006:**
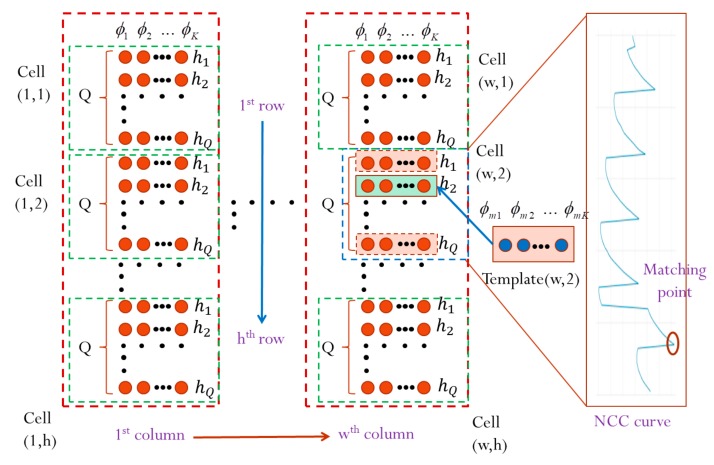
Illustration of the proposed absolute height calibration algorithm with every image pixel.

**Figure 7 sensors-19-04683-f007:**
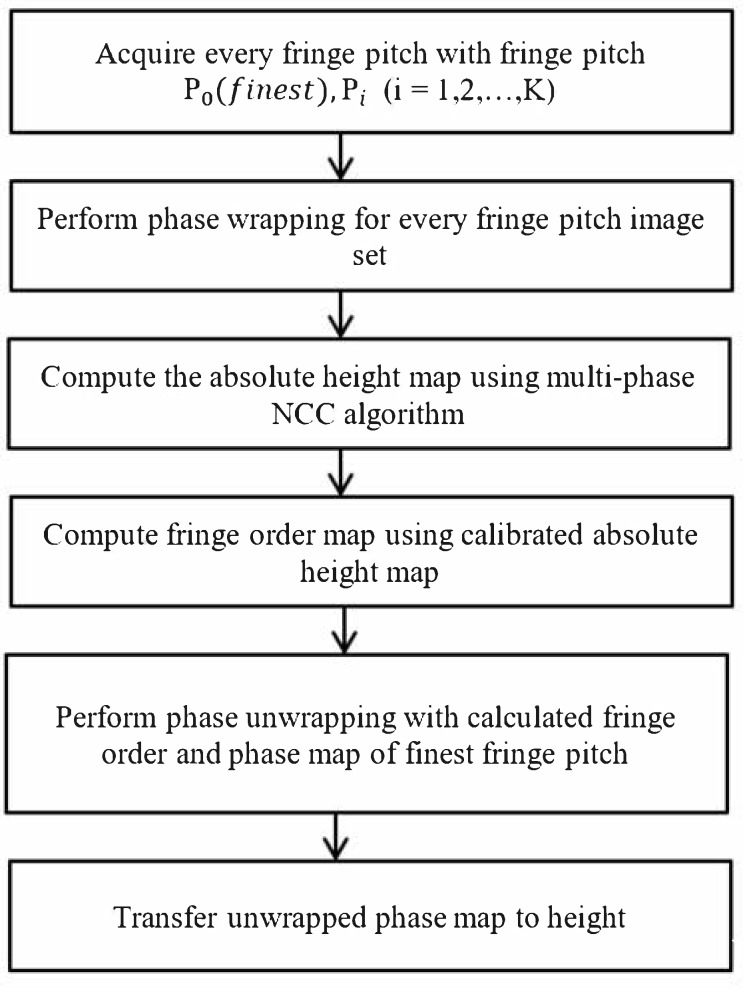
Flowchart of precise absolute depth measurement using the multiphase normalized cross-correlation (NCC) algorithm.

**Figure 8 sensors-19-04683-f008:**
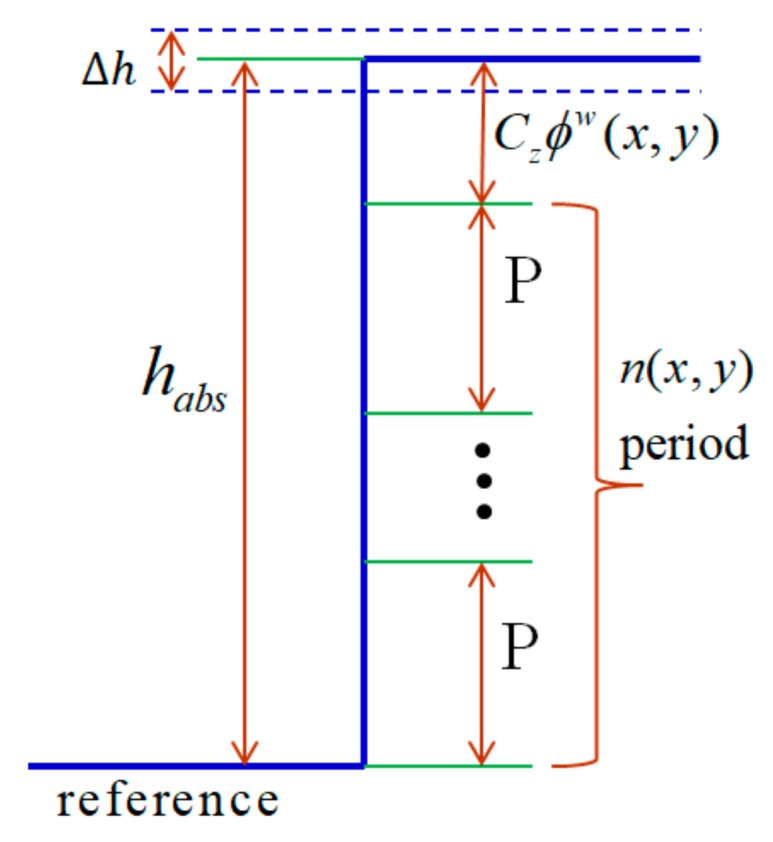
Illustration of fringe-order calculation using calibrated absolute height.

**Figure 9 sensors-19-04683-f009:**
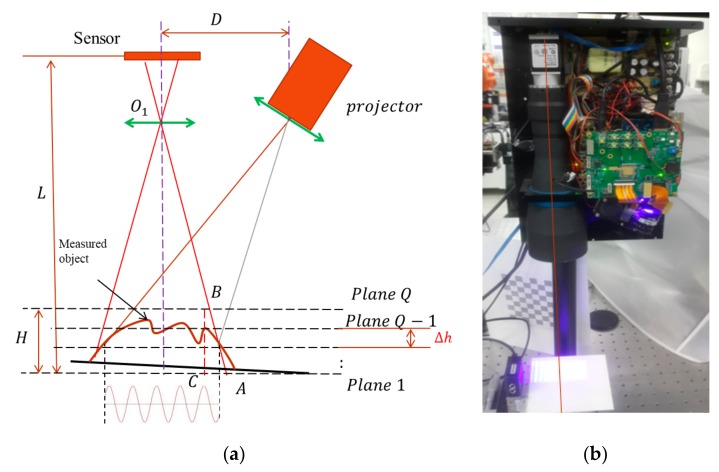
(**a**) Schematic diagram of a multifrequency phase-shifting (MFPS) system setup and (**b**) a real measurement system.

**Figure 10 sensors-19-04683-f010:**
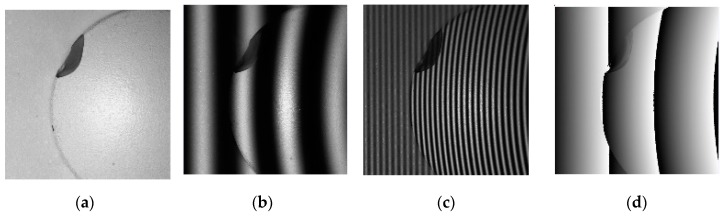
Captured image and its wrapped phase map with different projected patterns on semisphere target: (**a**) uniform light; (**b**) 121-pixel sinusoidal fringe pitch; (**c**) 12-pixel sinusoidal fringe pitch; (**d**) wrapped phase of (**b**); (**e**) wrapped phase of (**c**); (**f**) absolute height map calculated using proposed algorithm; (**g**) 3D profile; and (**h**) height cross-section of row 512 of (**f**).

**Figure 11 sensors-19-04683-f011:**
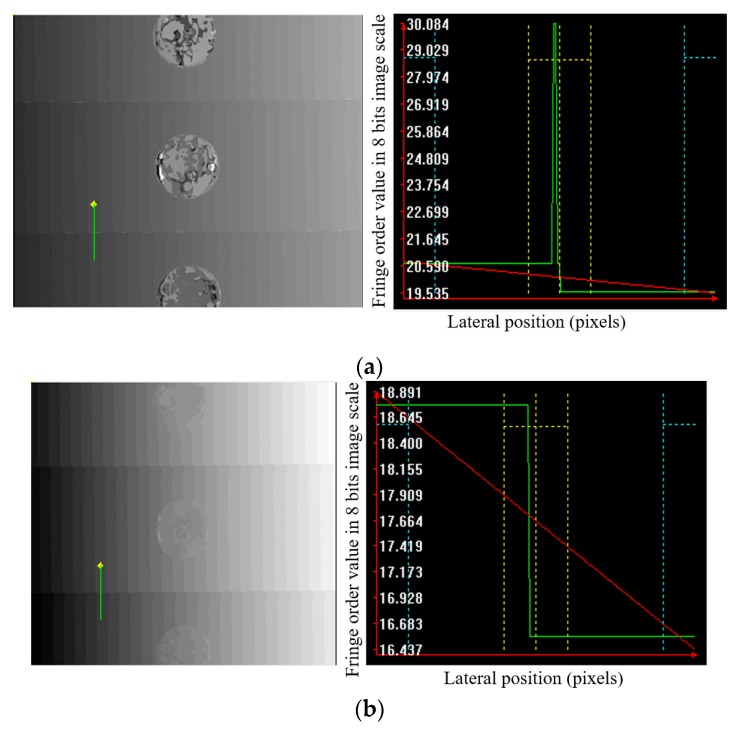
Comparison of fringe-order determination and phase ambiguity for a step-height target measured using (**a**) the MFPS algorithm and (**b**) the proposed fringe-order calculation algorithm.

**Figure 12 sensors-19-04683-f012:**
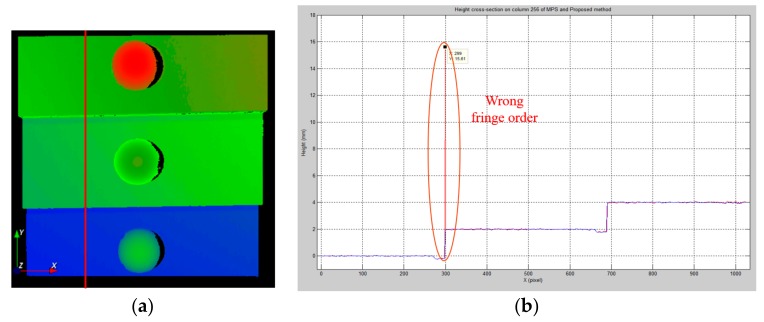
Measurement results for the 2-mm step-height target in Figure 11: (**a**) 3D profile; (**b**) height cross-sections on column 256 for both MFPS (red) and NCCMFPS (blue).

**Figure 13 sensors-19-04683-f013:**
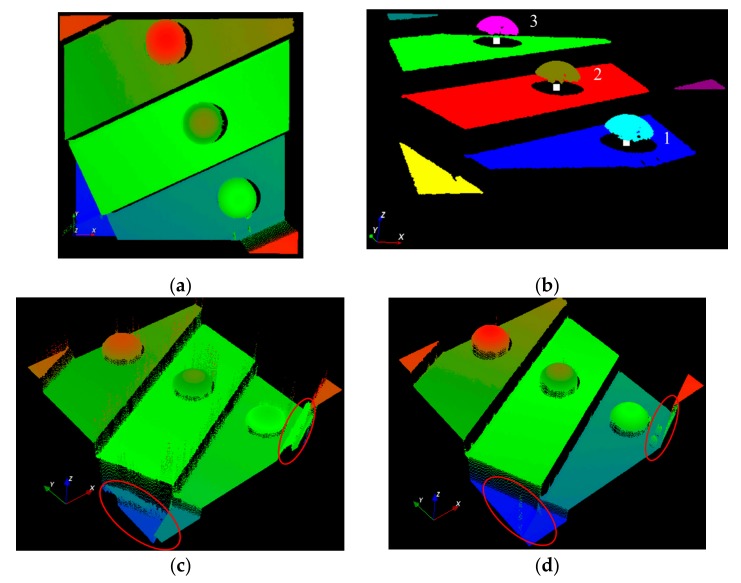
Measured 3D profile of step-height target: (**a**) 3D profile; (**b**) detected ball centers in (**a**); (**c**) 3D profile measured using MFPS; and (**d**) 3D profile measured using NCCMFPS.

**Figure 14 sensors-19-04683-f014:**
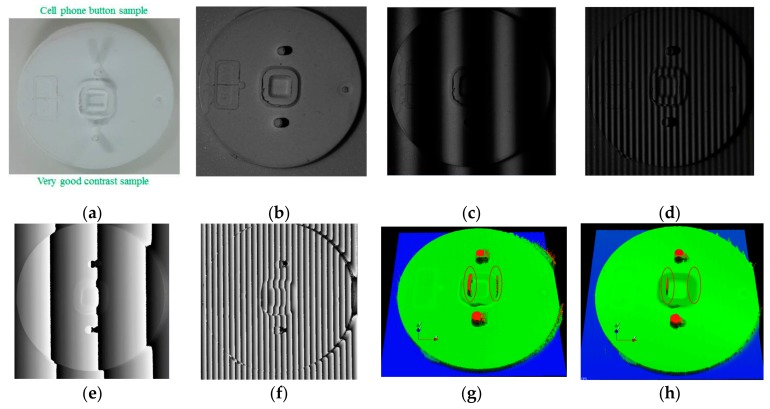
Measurement results of cell phone button target: (**a**) digital image of cell phone button target; (**b**) captured image without fringe; (**c**) captured image with fringe pitch 121; (**d**) captured image with fringe pitch 12; (**e**) wrapped phase map of fringe pitch 121; (**f**) wrapped phase map of fringe pitch 12; (**g**) 3D profile measured using MFPS; (**h**) 3D profile measured using NCCMFPS.

**Figure 15 sensors-19-04683-f015:**
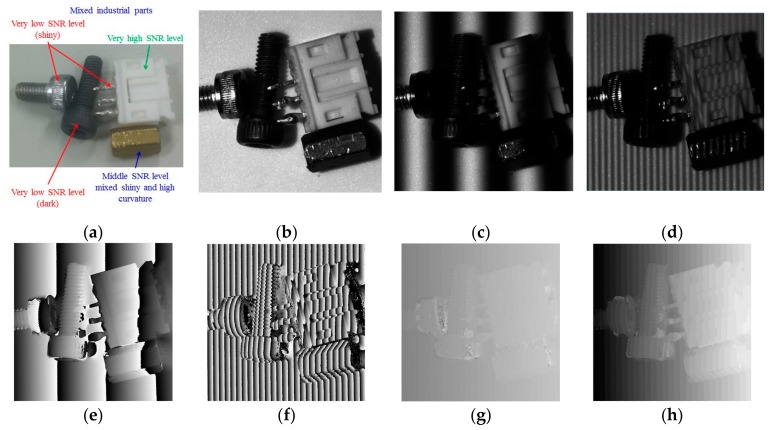
Measurement results of mixed industrial parts: (**a**) mixed industrial parts; (**b**) captured image without fringe; (**c**) captured image with fringe pitch 121; (**d**) captured image with fringe pitch 12; (**e**) wrapped phase map of fringe pitch 121; (**f**) wrapped phase map of fringe pitch 12; (**g**) fringe-order map calculated using MFPS; (**h**) fringe-order map extracted using NCCMFPS; (**i**) 3D profile measured using MFPS; (**j**) 3D profile measured using NCCMFPS.

**Table 1 sensors-19-04683-t001:** Results for step-height target measured using different approaches.

Absolute Height	Mitutoyo CMM (mm)	MFPS (mm)	NCCMFPS (mm)	Error of MFPS (mm)	Error of NCCMFPS (mm)
Pixel 299	2.020	15.61	2.010	13.59	0.010

**Table 2 sensors-19-04683-t002:** Comparison of distances between ball centers measured using NCCMFPS and the Mitutoyo CMM.

	Euclidean Distance (between)	Mitutoyo CMM (mm)	NCCMFPS (mm)	Error (mm)
Absolute distance (mm)	Balls 1 and 2	8.2608	8.2549	0.0058
	Balls 2 and 3	8.2592	8.2638	0.0046
Standard deviation (mm)	Balls 1 and 2	0.001	0.004	
	Balls 2 and 3	0.001	0.005	

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
