# Peer review of "Absolute Depth Measurement Using Multiphase Normalized Cross-Correlation for Precise Optical Profilometry"

_sensors, 2019, doi:10.3390/s19214683_

Round 1

Reviewer 1 Report

This manuscript proposes yet another temporal multiple frequency profilometry technique. I said yet another because this topic has already been exhaustively investigated and it is difficult for me to find something new and more efficient than the current state of the art in temporal multi-frequency profilometry.

To begin with, the authors explain too much (too many text lines) the well known limitations of spatial phase unwrapping. These pages can be reduced to just a paragraph without sacrificing the manuscript; this manuscript is only related with temporal phase unwrapping. So this manuscript is unnecessarily long.

Another major drawback of this manuscript is that (except for Ref [16] , 2016) all references are many years old for a research article.

Specific comments

(1) The first "new" (no reference is given) equation in this manuscript is Eq. (12). However this (equivalent) equation was published before by  Zuo, Huang, Zhang, Chen, Asundi, A. Opt. Lasers Eng. 2016. (Eq. (6) of Zuo et al. manuscript).

(2) The "new" section 3, of this manuscript "Proposed NCC-based methodology" has a major error (Eq.15, line 230). Since (2015) (Ref. [4] below) it was mathematically demonstrated that using the 2-sensitivities, temporal phase unwrapping, the phase-errors in the low frequency measurement do not propagate towards the higher frequency phase measurement. This was demonstrated in section 4 (Eqs (5)-(10)) of [4] (2015) referenced herein. This sole erroneous equation/statement invalidates the whole paper, I think.

In conclusion.

I really do not think that this manuscript contains anything new in temporal phase unwrapping for multi-frequency profilometry that has not already been published in the included references, and those 4 newer references that I have appended in this review. However I would reconsidered re-revising this manuscript if the authors can mathematically prove that their proposed method is more efficient in any way, to the given references [1]-[4] listed below in this revision.

REFERENCES for this reviewing

[1] Testing aspherics using two-wavelength holography, Appl Opt, 10 (1971), p. 2113
[2] Multi-frequency projected fringe profilometry for measuring objects with large depth discontinuities, Opt Commun, 288 (2013).
[3] Super-sensitive two-wavelength fringe projection
profilometry with 2-sensitivities temporal unwrapping., Optics and Lasers in Engineering 106, 68-74 (2018).
[4] Recovery of absolute height from wrapped phase maps for fringe projection profilometry
Opt Express, 22, p. 16819,(2014).
[5] Dual-sensitivity profilometry with defocused projection of
binary fringes, Applied optics 56 (28), 7985-7989 (2017).
[6] Dual wavelength digital holography for improving the measurement accuracy, Proc SPIE – Int Soc Opt Eng, 8769 (2013).
[7] Profilometry of three-dimensional discontinuous solids by combining two-steps temporal phase unwrapping, co-phased profilometry and phase-shifting interferometry , Optics and Lasers in Engineering 87, 75-82 (2016).
[8] Improvement of measurement accuracy in digital holographic microscopy by using dual-wavelength technique, J Micro/Nanolithogr, MEMS, MOEMS, 14 (2015).
[9] Temporal phase-unwrapping of static surfaces with 2-sensitivity fringe-patterns, Optics express 23 (12), 15806-15815 (2015).
[10] Improved phase retrieval method of dual-wavelength interferometry based on a shorter synthetic-wavelength, Opt Express, 25 (2017).

Reviewer 2 Report

Authors describe a new absolute height and fringe order calculation developed from multi-phase NCC. Results show a minimization of fringe ordering error in MFPS. The proposed methodology aims at eliminating the error propagation in estimating the difference between measured and a priori data.

Authors, with their novel approach to the minimization of the error propagation, propose to overcome actual limitations (I am not able to understand if such limitations are actually recognized within the research community) and improve the obtainable results.
The paper is clear, yet exaggeratedly long and detailed in the mathematical formulation of both the state of the art and the proposed approach.
The experimental setup even though limited to few samples, is well designed and conveys all the specific test needed to evaluate the overall system performance.

I report some minor considerations:

Figure 2; cite Goldstein

Figure 9: hidden words

Rephrase rows 134-136

typos: rows 64, 101, 164, 194,195, 321, 349, 440

Reviewer 3 Report

In this manuscript, the author proposed a new MFPS algorithm for absolute depth measurement.  With normalized cross-correlation, the fringe order mismatch problem can be addressed, and the proposed algorithm shows better accuracy compared to the traditional MFPS methods.  Overall all, this manuscript is in a good shape and I recommend minor revision for this paper by considering the following points:

The review section 2 has too much details. These methods are well established and reported in existing literatures. These contents can be condensed into a table or shorter section. What are the image resolutions in the experiment setup? Is the developed NCCMFPS method will depend on the camera resolution and DOF? Figure 9 is cropped. Check the format. The manuscript should be reformatted carefully. Some reference is missing (i.e. line 440). Some tracked changes are still remain in the sentences.   The claim of 47% accuracy improvement in height measurement looks not very rigorous. Is the accuracy improvement will change with object height, camera resolution, fringe pitch size, etc.?

Round 2

Reviewer 1 Report

I do not recommend further improvements